

# Two-stage multi-objective evolutionary algorithm for overlapping community discovery

Lei Cai[1,2], Jincheng Zhou[1] and Dan Wang[3]

[1] Key Laboratory of Complex Systems and Intelligent Optimization of Guizhou Province, School of Computer and Information, Qiannan Normal University for Nationalities, Duyun, Guizhou, China
[2] State Key Laboratory of Public Big Data, College of Computer Science and Technology, Guizhou University, Guiyang, China
[3] School of Mathematics and Statistics, Qiannan Normal University for Nationalities, Duyun, Guizhou, China

Corresponding author
Jincheng Zhou, zjc81@sgmtu.edu.cn

## ABSTRACT

As one of the essential topological structures in complex networks, community structure has significant theoretical and application value and has attracted the attention of researchers in many fields. In a social network, individuals may belong to different communities simultaneously, such as a workgroup and a hobby group. Therefore, overlapping community discovery can help us understand and model the network structure of these multiple relationships more accurately. This article proposes a two-stage multi-objective evolutionary algorithm for overlapping community discovery problem. First, using the initialization method to divide the central node based on node degree, combined with the cross-mutation evolution strategy of the genome matrix, the first stage of non-overlapping community division is completed on the decomposition-based multi-objective optimization framework. Then, based on the result set of the first stage, appropriate nodes are selected from each individual's community as the central node of the initial population in the second stage, and the fuzzy threshold is optimized through the fuzzy clustering method based on evolutionary calculation and the feedback model, to find reasonable overlapping nodes. Finally, tests are conducted on synthetic datasets and real datasets. The statistical results demonstrate that compared with other representative algorithms, this algorithm performs optimally on test instances and has better results.

# INTRODUCTION

Complex networks are usually abstracted from practical applications in the real world. It is a network structure composed of a large number of interconnected nodes. Each node represents an individual in the actual application, and the connected edges represent the relationship between individuals. In the real world, biological networks (*Rahiminejad, Maurya & Subramaniam, 2019*), scientific collaborations networks (*Bütün & Kaya, 2019*), social networks (*Tajeuna, Bouguessa & Wang, 2018*; *Abd Al-Azim et al., 2022*), *etc.*, are all

typical representatives of complex networks. Studies have shown that these networks have characteristics such as functions, cooperation models, and social relationships in their respective fields. Similarly, they also share common characteristics, such as community structure (*Newman & Reinert, 2016*; *Fortunato & Hric, 2016*), scale-free (*Barabási, Albert & Jeong, 2000*), and small-world (*Watts & Strogatz, 1998*). These characteristics make complex networks a powerful tool for studying and simulating various systems.

Community structure refers to several groups formed by close connections between nodes in the network, in which nodes are relatively tightly connected to other nodes within the group, but relatively sparsely connected to other nodes outside the group. Community discovery is the process of identifying community structures in a network, to divide the network into different structures based on the sparsity of connections in the network (*Gong et al., 2013*). From the perspective of whether the communities to which network nodes belong overlap, community discovery problems can be divided into non-overlapping community discovery problems and overlapping community discovery problems. In traditional non-overlapping community discovery, nodes are assigned to a single community, while overlapping community discovery allows nodes to belong to multiple communities at the same time. In the real world, the relationships between nodes are often complex and diverse. For example, in a scientist partnership, some scientists may be involved in different fields and directions; in a social network, a person may be a member of both a workgroup and a hobby group. Therefore, overlapping community discovery can help us understand and simulate the network structure of these multiple relationship more accurately.

Due to the complexity and diversity of overlapping communities, their discovery and analysis also face a series of challenges, including how to efficiently discover overlapping communities and how to improve the quality of overlapping communities. Although existing overlapping community discovery algorithms have made certain progress, there is still room for improvement. In the process of searching communities, these algorithms are often easily affected by local information and sometimes suffer from overfitting. In addition, some algorithms are sensitive to the node selection and initial conditions of the initial population, which may lead to instability in the results. In addition, algorithms based on evolutionary computation usually only consider the target value of the current evolutionary stage and ignore the previous historical information, which may lead to the loss of valuable information. To address these issues existing in the current overlapping community discovery algorithm and improve the accuracy and effectiveness of overlapping node classification in the overlapping community discovery algorithm, this article uses the relevant knowledge of overlapping community discovery to propose a two-stage multi-objective evolutionary overlapping community discovery algorithm. The main contributions of this article are as follows:

- to obtain better non-overlapping communities and provide a suitable initial population for the second stage, this article adopts a population initialization strategy based on central nodes in the first stage. This strategy combines the maximum node degree and the random generation of central nodes to improve the diversity and coverage of the

population. At the same time, by using a method based on multi-objective evolution to optimize non-overlapping communities, it aims to comprehensively consider multiple optimization objectives of the community to better discover high-quality community structures;

- to obtain better overlapping communities, in the second stage, based on the first stage, this article randomly selects a node from each community in the divided non-overlapping communities, whose node degree is greater than the average node degree of the community, and Makes it the central node of this community. In this way, we gradually find the central nodes of all communities and form an initial population. In addition, this article adopts an information feedback model and uses historical information to optimize the fuzzy threshold at the current evolutionary stage to more accurately identify overlapping nodes and thereby obtain more appropriate overlapping communities.

The remainder of this article is presented below. We review related work in "Related Work". "Description of the proposed algorithm" introduces the working principle of this work's two-stage optimization method in overlapping community discovery. "Experiment" conducts experiments on synthetic and real networks to test the effectiveness and authenticity of the proposed algorithm. A summary of the proposed algorithm and a discussion of possible future improvements are described in "Conclusion and Future Work".

## RELATED WORK

Currently, numerous valuable methods for overlapping community detection have been proposed by researchers. These include approaches utilizing label propagation, local expansion, fuzzy clustering, and evolutionary algorithms. The following focuses on the development process of relevant methods.

(1) Label-based propagation method

The overlapping community discovery algorithm based on label propagation is an extension of *Raghavan, Albert & Kumara (2007)* in label propagation algorithm (LPA), and nodes with the same label form a community. In the overlapping variant of LPA, nodes are allowed to have multiple labels. *Gregory (2010)* proposed the community overlap propagation algorithm (COPRA) algorithm to detect overlapping communities in the network through label propagation. The algorithm introduced a new label structure and propagation step, and set overlap parameters to limit nodes that can belong to more than one community at the same time. The number of communities makes it difficult to select appropriate parameters to divide into appropriate overlapping communities. *Wu et al. (2012)* proposed a balanced multi-label propagation algorithm (BMLPA) for overlapping community detection based on the COPRA algorithm. This algorithm proposes a balanced belonging coefficient label update strategy and a new initialization vertex label, which can improve the quality and stability of community detection results. *Sheng et al. (2019)* proposed the preferential learning and label propagation algorithm (PLPA) algorithm. The

preference learning mechanism based on learning behavior and information interaction uses the degree of preference for neighbor nodes to select learning targets to update node labels, but it has strong randomness and low accuracy. *Yan et al. (2023)* proposed a Fast Label Propagation Algorithm (FLPA), which uses graph compression technology to reduce the size of the network and combines the node influence calculation method and alpha path similarity to accurately control label weights. This type of method is easily affected by local information and may cause overfitting in some cases.

(2) The method based on local expansion

*Lancichinetti, Fortunato & Kertész (2009)* proposed a method based on local optimization of fitness function (LFM). This algorithm randomly selects core nodes as the initial community and then selects the one with the highest fitness value by calculating the fitness value of each node. nodes and add them to the community, and continue to expand until all nodes are divided into a certain community. *Jia, Du & Liu (2021)* proposed a locally optimized overlapping community discovery algorithm integrating attribute features. This method obtains core nodes by improving the density peak fast search algorithm and combining node attributes and structural information, using core nodes and non-core nodes. The objective function established between nodes optimizes the discovery of communities and obtains better overlapping communities by merging communities with high overlap. *Zhuo et al. (2024)* proposed an expansion method with adjusted overlap (ECOCD), which uses non-negative matrix decomposition to obtain disjoint communities as core communities and then randomly selects a community for expansion and contraction. The final overlapping community is obtained when the overlap of all communities reaches a predefined threshold.

This type of method is sensitive to the node selection and initial conditions of the initial population, and can easily lead to instability in the results.

(3) The method based on fuzzy clustering

The fuzzy clustering method is considered an effective technique for detecting overlapping communities because it allows nodes to be assigned to multiple clusters simultaneously, thus satisfying the overlap between multiple communities. *Zhang, Wang & Zhang (2007)* proposed a method to identify overlapping communities in complex networks based on fuzzy concepts. This method combines fuzzy clustering with generalized modularization and spectral mapping and uses fuzzy c-means clustering of Euclidean distance. Class methods cluster data nodes. *Lv, Yang & Yang (2013)* combined spectrograms with fuzzy set theory and proposed a fuzzy spectrum-based FSC algorithm for detecting overlapping communities. This method uses membership degrees to allocate ownership rights of network nodes. *Tao et al. (2018)* proposed a new community discovery method with the fuzzy density peak clustering algorithm (CDFDPC). This method uses the shortest path and node similarity to design a distance matrix, adaptively selects thresholds and core communities, and combines fuzzy clustering theory to determine the degree of membership. The nodes of the matrix are allocated to get overlapping nodes.

*Jokar, Mosleh & Kheyrandish (2022)* combined the method of balanced link density label propagation and fuzzy theory. This method is divided into three stages. First, the edge weight is calculated based on the balanced density of edges in the network, and secondly, the preprocessed weight is used to achieve a stable label propagation technology, and finally the nodes of the community through fuzzy membership.

Although this type of method can effectively detect overlapping communities, it requires preprocessing prior knowledge to ensure the accuracy of the algorithm. Therefore, it may be challenging for the algorithm to select appropriate parameters for different network structures and data characteristics, and it needs to be adjusted empirically.

(4) Fuzzy clustering method based on evolutionary computation

Evolutionary computing has an advantage in solving multi-objective optimization problems. It is a random search method that can strongly search for global optimal solutions and can effectively solve the dilemma of local optimal algorithms. *Tian, Yang & Zhang (2019)* proposed an overlapping community detection fuzzy method based on evolutionary multi-objective optimization (EMOFM). This method uses a multi-objective evolutionary algorithm to optimize the community center. First, the center nodes of non-overlapping communities are optimized to obtain a more accurate result. center node, and secondly find overlapping nodes by optimizing the fuzzy threshold method. To improve the performance of the above method, two optimized population initialization strategies are also proposed. *Shang et al. (2022)* proposed a method based on a similarity matrix to initialize the central node. In the first stage, this method uses a similarity threshold to determine similarity communities to find the central nodes of non-overlapping communities. At the same time, the method of optimizing fuzzy thresholds in the second stage of *Tian, Yang & Zhang (2019)* is used to find overlapping nodes. Finally, the correction strategy is used to modify the nodes that may be unreasonably divided.

Although the above research work can effectively discover overlapping nodes, it relies on the selection of central nodes in non-overlapping communities during the optimization process. How to determine better central nodes can still be further improved. In addition, discovering overlapping nodes depends on fuzzy thresholds. During the evolution process, the above method only considers the fuzzy thresholds at the current evolutionary stage and ignores previous historical information. To address these issues, we adopt a two-stage approach to get a better initial population, use historical threshold information to find more accurate overlapping nodes and correct inaccurate nodes through node correction strategies.

## Description of the proposed algorithm

Figure 1 shows the optimization process for overlapping community discovery. First, a test network is given, and after the first stage of non-overlapping community discovery, a better non-overlapping community structure is obtained. Then, the obtained non-overlapping community structure is put through the second stage of overlapping community discovery, after which an accurate overlapping community structure can be obtained. Next, we will focus on the realization process of the two phases.

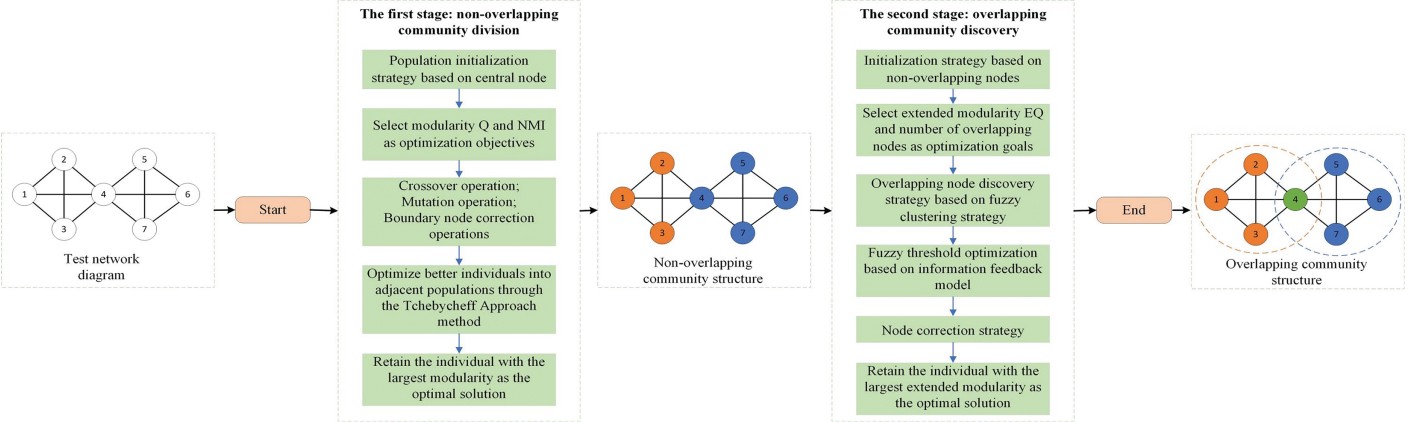

**Figure 1  Overlay community discovery optimization process diagram.**

## The first stage: non-overlapping community division

At this stage, to obtain higher-quality non-overlapping community structures, this article uses modularity and normalized mutual information as optimization objectives. We will introduce these two objectives in detail in the experimental section. Two objectives are optimized through a decomposition-based multi-objective optimization framework to obtain high-quality non-overlapping communities. For non-overlapping community discovery algorithms, it is mainly divided into an initialization part and an evolution part. The initialization part adopts the population initialization strategy based on the central node. In the evolution part, this article performs cross-mutation on individual genes and optimizes individuals by combining the boundary node occupancy allocation method (*Cai, Zhou & Wang, 2023*). Finally, by comparing all the offspring produced by parents, modules are selected from them. The individual with the highest degree is regarded as the optimal solution. The specific implementation is shown in Algorithm 1.

(1) Initialization strategy based on central node

   In the first stage of the population initialization strategy, this article uses an initialization strategy based on the central node to create initial individuals. First, in a real community, the node at the center of the community is likely to be more closely connected to other nodes. The node metric usually represents the number of connections between the node and other nodes. Therefore, the higher the node degree, it means that the node is in the network. With more connections, a node is more likely to become a community center node. Specific steps are as follows:

1) First, calculate the degree of each node in the network, and select the node with the largest degree as the initial central node.

2) Mark the initial central node as the central node of the initial community, and assign all nodes directly connected to the central node to the initial community to form the first initial community.

| **Algorithm 1 Non-overlapping community discovery.** |
| --- |
| **Input:** undivided network structure, adjacency matrix *adjacent*, population number *pop_size*, maximum number of iterations *maxgen* |
| **Output:** non-overlapping community structure |
| 1.    Initialize weight vector and neighborhood information |
| 2.    Population initialization strategy based on central node |
| 3.    Calculate the modularity Q and normalized mutual information NMI of the population |
| 4.    **for** *gen* =1: *maxgen* **do** |
| 5.        **for** *pop_id* =1: *pop_size* **do** |
| 6.            Randomly select two different individuals from the population |
| 7.            Perform a crossover operation to generate a new individual *y* |
| 8.            **if** rand(1)<0.3 **then** |
| 9.                Perform mutation operation on individual *y* |
| 10.           **else** |
| 11.               Correction of nodes in individual *y* |
| 12.           **end** |
| 13.           Optimize better individuals into adjacent populations through the Tchebycheff Approach method |
| 14.       **end** |
| 15. **end** |
|      Output non-overlapping community structure |

3) Among the remaining unassigned nodes, find the node with the largest degree as the center node of the next community.

4) For each newly selected central node, allocate its directly connected unallocated nodes to the community of the central node.

5) Repeat steps 3 and 4 until all nodes are assigned to the community of a certain central node.

Through this method, a well-constructed network community structure can be initially obtained. Each central node and its directly connected neighbor nodes constitute a community of the network. In addition, to improve the diversity of the initial population, only half of the individuals use the above-mentioned selection strategy of maximum node degree as the central node, and the remaining individuals are generated by randomly selecting nodes as the central node.

(2) Multi-objective evolution strategy based on MOEA/D

To search efficiently and accurately to obtain high-quality community structures, this article uses a multi-objective evolutionary algorithm framework based on decomposition to optimize the two objectives of modularity and normalized mutual information. After using the two-way crossover mutation method of the genome matrix during the evolution process, reasonable communities can be searched more quickly and effectively, and the

diversity of the population can be improved while avoiding falling into local optimality during the search process. In addition, the node correction strategy will be Section "Experiment" introduces it in detail.

Before performing the two-way crossover mutation operation, we convert the community structure into the form of an adjacency matrix. The intersection values of rows and columns in the adjacency matrix represent the relationship between two nodes. Among them, 1 indicates that there is a connection between the two nodes, 0 indicates that there is no connection between the two nodes, and −1 indicates that the two nodes are in the same community structure.

1) Two-way crossover mutation strategy

To form a new population, two individuals are randomly selected as parents and a crossover operation is performed to generate new individuals. The crossover mutation operation occurs directly in the individual's genome, by randomly selecting different numbers of nodes in the parent generation and exchanging the genetic attribute values of the same nodes in different parents to generate new individuals. In addition, to prevent the generated individuals from falling into the local optimum, their genetic attributes are mutated with a certain probability. The mutation operation also randomly selects different numbers of nodes and inverts the genetic attributes of the nodes to generate new individuals and increase the population diversity. For example, as shown in Fig. 2, two individuals (*a*) and (*b*) are randomly selected for crossover operation. Assume that nodes 3 and 4 are randomly selected, and the genetic attribute values of nodes 3 and 4 in (*b*) are assigned to (*a*), the gene attribute values of the remaining nodes remain unchanged, and a new individual is generated (*c*). It can be found that (*b*) and (*c*) are the same, so mutation operations will continue to be performed on them with a certain probability to avoid duplication of the generated individuals. It is also assumed that nodes 3 and 4 are randomly selected for mutation operation, and the genetic attribute values of nodes 3 and 4 in (*c*) are inverted, and finally a new individual (*d*) is generated, thereby completing an evolutionary process.

2) Individual choice

Individual selection retains the optimal solution in a manner that eliminates poorer candidate solutions during the evolutionary iteration process. The Tchebycheff Approach method is used to calculate the weighted maximum deviation of modularity and normalized mutual information and retain the value with the smallest deviation to the neighborhood information to complete the individual update. After completing the evolution for the prescribed number of iterations, a group of communities with the same size and higher quality as the initial population can be obtained, making the initial population in the second stage closer to the real community.

## The second stage: overlapping community discovery
The second stage is the process of finding overlapping nodes based on the non-overlapping communities in the first stage. On the decomposition-based multi-objective optimization

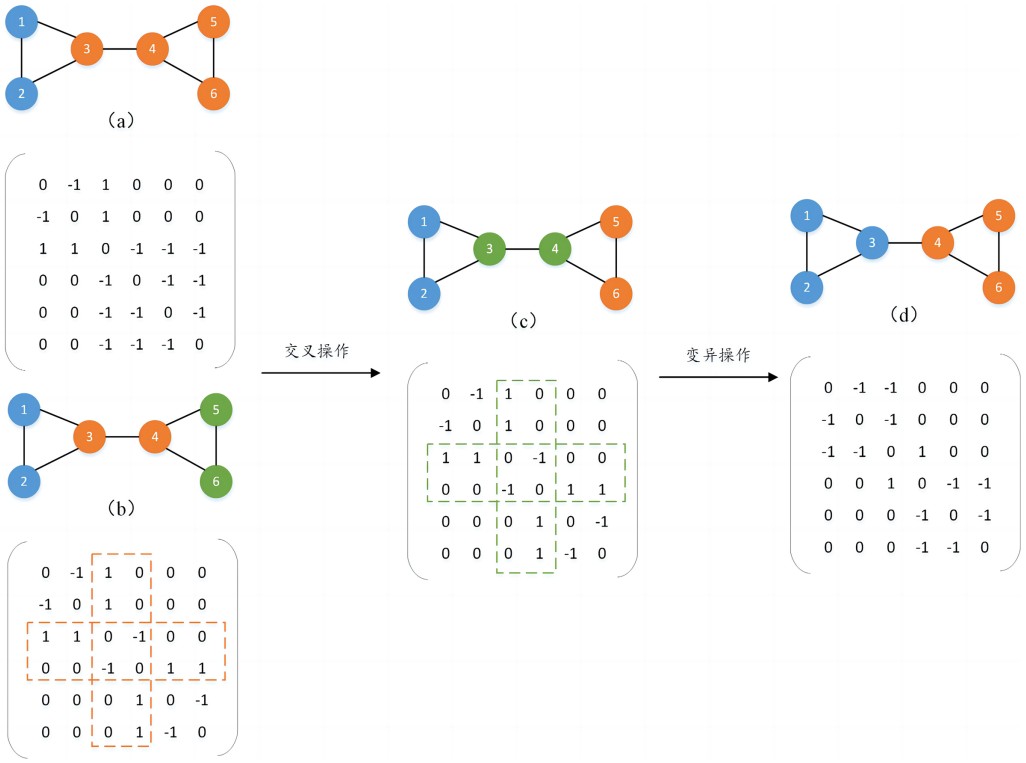

**Figure 2 Two-way crossover mutation process diagram.**

framework MOEA/D, the population is optimized with the extended modularity EQ and the number of overlaps as optimization objectives. The initialization population in the second stage benefits from the results of the first stage. The central node and non-central node of each community are found in the divided non-overlapping communities as the population initialization in the second stage. Inspired by *Tian, Yang & Zhang (2019)*, we used the fuzzy clustering idea to assign a fuzzy threshold to each node to control whether the node is overlapping. During the optimization process, we not only perform cross-mutation on the current individual to generate new individuals but also use the information feedback model (*Wang & Tan, 2017*) to guide the generation of subsequent individuals to make full use of the valuable information before iteration. The specific algorithm implementation is Algorithm 2.

(1) Initialization strategy based on non-overlapping nodes

In the previous stage, this article divided network nodes into non-overlapping communities. To obtain better overlapping community division results at this stage, a good initialization population is essential. Therefore, in this stage of population initialization, this article randomly selects a node from each community in non-overlapping communities from the node set with a node degree greater than the average node degree as the central node of the population, and its remaining nodes are non-central. In this way, we get a better initial population of overlapping communities. In addition, to diversify the

| Algorithm 2 | Overlapping community discovery. |
|---|---|

**Input:** Adjacency matrix *adjacent*, population number *pop_size*, maximum number of iterations *Gen*, community division in the previous stage *no_Community*

**Output:** populations of overlapping communities

1:    Based on the initial population of non-overlapping communities, the central node is obtained from each community as the initial population of overlapping communities.

2:    Calculate the target value EQ of the population and the number of overlapping nodes *Numoverlapping*, and arrange them in descending order.

    **for** *pop* =1: *pop_size* **do**

3:    A set of populations is generated for each individual for optimization.

4:    Overlapping node discovery strategy based on fuzzy clustering ideas.

5:    Initialization method based on MOEA/D.

6:    *j*=0;

7:    **for** *Gene* =1: *Generations* **do**

8:     *j*=*j*+1;

9:     Perform a crossover mutation operation on each population to generate a new population

10:    **for** *pop_id* = 1 : *pop_size* **do**

11:    **if** *j*>2 **then**

12:     Using fuzzy threshold based on information feedback model to optimize new population y.

13:    **end**

14:     Optimize better individuals into adjacent fields through the Tchebycheff Approach method.

15:    **end**

16: **end**

17: **end**

18: The non-dominated solutions in all output results are the divided overlapping community populations.

initial population and avoid a single initial population, this article adopts the above scheme for half of the initial population, and the remaining initial population randomly selects nodes from each community of non-overlapping communities as the initial central node.

(2) Overlapping node discovery strategy based on fuzzy clustering strategy

Fuzzy clustering is a clustering method that permits a sample to be associated with multiple clusters, facilitating its integration into the process of identifying overlapping nodes. During this process, the sets of central and non-central nodes identified from the initial population are denoted as illustrated in Eq. (1).

$$CN = \{CN_1, CN_2, \ldots, CN_n\}$$
$$NCN = \{NCN_1, NCN_2, \ldots, NCN_m\}$$

(1)

Among them, the sum of *n* and *m* is the total number of nodes, *CN* represents the central node-set, and *NCN* represents the non-central node set. The node number position corresponding to the central node is 1, and the non-central node is 0.

To effectively identify overlapping nodes, it is necessary to calculate the membership matrix between the central node and the non-central node and the fuzzy threshold of each node. Calculating the distance between the central node and the non-central node according to Formula (2) (*Bezdek, Ehrlich & Full, 1984*) can obtain each element in the membership matrix.

$$A_{ij} = \frac{1}{\sum_{k=1}^{|CN|} \frac{dis(NCN_i, CN_j)}{dis(NCN_i, CN_k)}^{\frac{2}{f-1}}} \tag{2}$$

Among them, $A_{ij}$ represents the membership degree between $NCN_i$ and $CN_j$, $dis(NCN_i, CN_j)$ represents the distance between the two nodes, $i \in [1, n], j \in [1, m]$, and $f$ is the parameter that controls fuzzy clustering. It is generally recommended to be 2. In the initial stage, each node will randomly generate a fuzzy threshold $r$. According to Formula (3), the appropriate fuzzy threshold $r_i'$ can be found to divide overlapping nodes.

$$r = (r_1, r_2, \ldots, r_n), r_i \in [0, 1]$$
$$r' = \min A_{il} + r_i \times (\max A_{il} - \min A_{il}) \tag{3}$$

where $r_i$ is the fuzzy threshold of the $i$th node, and $A_{il}$ represents all columns of the $i$th row of the membership matrix. By comparing the element values in the membership matrix with the fuzzy threshold, suitable overlapping nodes can be found. To better understand this process, a simple division example is given as shown in Fig. 3.

In the above legend, the central nodes are node 1 and node 6, represented by 1 in the vector, and the remaining nodes are non-central nodes, represented by 0; the fuzzy threshold $r$ is randomly generated, and each threshold corresponds to a non-central node at the same position. central node. The membership matrix $A_{ij}$ and the fuzzy threshold $r_i'$ are calculated according to Eqs. (2) and (3). By comparing the sizes of $r_i'$ and $A_{il}$, the node will preferentially choose to join the community with a $r_i'$ value less than or equal to $A_{ij}$. For example, $r_2' = 0.5417$, $A_{21} = 0.7083$, $A_{22} = 0.2917$, $r_2' < A_{21}$, so node 2 will choose to join community 1, and all nodes will be compared in the same way. After all nodes are selected, they can be converted into overlapping communities.

(3) Fuzzy threshold optimization based on information feedback model

The information feedback model uses previous information to guide the subsequent update process in a meta-heuristic algorithm. In the process of overlapping community discovery, more accurate overlapping nodes can be obtained by rationally using previous information to optimize the fuzzy threshold. In the process of selecting previous individual information, there are two methods: random selection and fixed selection. To increase the diversity of the population, this article uses the random selection method to select individuals. At the same time, to reduce the complexity of the algorithm, in this model only the previous historical information of the current update process is selected to guide the current update status. The specific formulas (*Wang & Tan, 2017*) are as follows Eqs. (4)–(6).

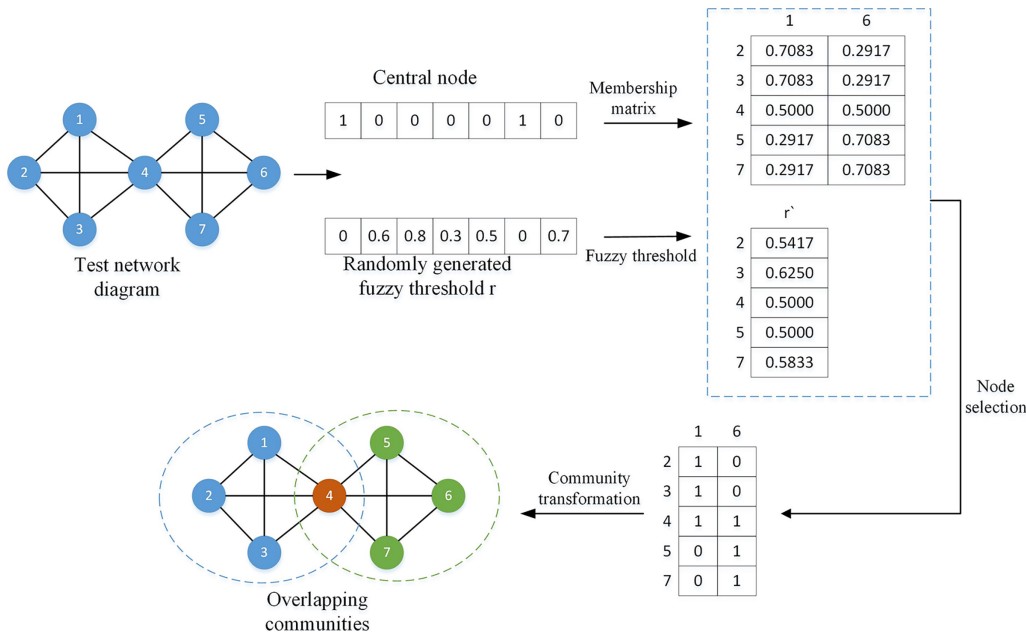

**Figure 3 Overlapping node division diagram.**

$$r_i^{t+1} = \theta_1 r_{rand}^t + \theta_2 y_i^{t+1} \tag{4}$$

$$\theta_1 = \frac{F^{t+1}}{f_{rand}^t + F^{t+1}} \tag{5}$$

$$\theta_2 = \frac{f_{rand}^t}{f_{rand}^t + F^{t+1}} \tag{6}$$

Among them, $r_{rand}^t$ is the randomly selected fuzzy threshold for the $t$ iteration of an individual, and the corresponding fitness value is $f_{rand}^t$; $y_i^{t+1}$ is the individual obtained by the current algorithm iteration, and the corresponding fitness value is $F^{t+1}$.

(4) Node correction strategy

Due to the randomness of the evolutionary algorithm, some nodes may be mistakenly assigned to unreasonable communities in the overlapping community population obtained according to the above method. To reduce the influence of algorithm randomness, this article proposes a correction strategy to reallocate overlapping nodes and non-overlapping nodes. Based on the idea of boundary node occupancy allocation, the occupancy of each node in all neighboring communities is calculated, and unreasonable nodes are allocated to communities with large occupancy. The specific occupancy formula is as shown in Eq. (7).

$$f_C^k = \frac{d_C^k}{\sum\limits_{i,j \in C} e_{ij}} \tag{7}$$

Among them, $d_C^k$ represents the degree of the $k$th node in community $C$, $k \in [1, n]$, and $e_{ij}$ represents the edge connecting nodes in community $C$. If there is an edge between nodes, it is 1, otherwise it is 0.

In the correction strategy, the correction of nodes is generally divided into three situations. First, when the occupancy rate of the community where the node is located is greater than the occupancy rate of the node's neighboring community, the node is correct; second, when the occupancy rate of the community where the node is located is equal to If the node is a non-overlapping node, then the node is converted into an overlapping node; thirdly, when the occupancy of the community where the node is located is less than the occupancy of the node's neighborhood community, the node is corrected to the maximum share of the community.

## EXPERIMENT

To evaluate the effectiveness of the overlapping community discovery algorithm proposed in this article, the proposed algorithm is compared with the MRMOEA algorithm (*Zhang et al., 2017*), the EMOFM-DK algorithm, and the CNLLP algorithm (*Zhang, Shang & Jiao, 2023*) on synthetic datasets and real datasets. In addition, to evaluate the effectiveness of the first-stage non-overlapping community discovery algorithm, we compared the proposed algorithm's first-stage non-overlapping community discovery modularity value with the following three algorithms on seven real data sets, including the first stage of the EMOFM-DK algorithm, the first-time step of the DECS algorithm, and the CNLLP algorithm. For each baseline algorithm, we adopt the parameters suggested in their articles. For experimental fairness, we set the population size of all algorithms to 100.

The parameter Settings in this article are shown in Table 1, where pop_size indicates the population size, Gen indicates the maximum number of iterations in the second stage, maxgen indicates the maximum number of iterations in the first stage, and N_neighbor indicates the size of the neighbourhood setting.

The running environment of this experiment is Windows 11 operating system, AMD Ryzen 5 5600H with Radeon Graphics 3.30 GHz and 16.0 GB memory. The software uses Matlab2019.

### Evaluation indicators

(1) Extended modularity function EQ

The extended modularity function EQ (*Ma et al., 2021*) is an extension of traditional modularity and is used to evaluate the performance of overlapping community divisions. By maximizing the extended modularity, the algorithm can more accurately identify multiple overlapping community structures existing in the network, as shown in Eq. (8).

$$EQ = \frac{1}{2m} \sum_{i=1}^{t} \sum_{v \in C_i, w \in C_i} \frac{1}{O_v O_w} \left[ A_{vw} - \frac{k_v k_w}{2m} \right] \tag{8}$$

**Table 1  The parameter setting in experiments.**

| Parameter | Value |
| --- | --- |
| pop_size | 100 |
| Gen | 100 |
| maxgen | 20 |
| N_neighbor | 5 |

Among them, $t$ represents the number of communities, $C_i$ represents the $i$th community, $O_v$ and $O_w$ represents the number of communities to which nodes $v$ and $w$ respectively belong. When both $O_v$ and $O_w$ only belong to one community, the extended modularity will return to the traditional modularity.

(2) Generalized normalized mutual information

Generalized normalized mutual information (*Lancichinetti, Fortunato & Kertész, 2009*) refers to comparing the community structure detected by the algorithm with the real community of the ground truth, and can only be used in networks where real community divisions exist. Its range $gNMI(A, B) \in [0, 1]$, the larger the calculated value, indicates that the community structure detected by the algorithm is closer to the ground truth community structure. As shown in Eq. (9).

$$gNMI(A, B) = \frac{-2 \sum_{i=1}^{m^A} \sum_{j=1}^{m^B} L_{ij} \log(L_{ij} N / L_{i.} L_{.j})}{\sum_{i=1}^{m^A} L_{i.} \log(L_{i.}/N) + \sum_{j=1}^{m^B} L_{.j} \log(L_{.j}/N)} \tag{9}$$

where $m^A$ and $m^B$ represent the number of community $A$ and community $B$ respectively, $L_{ij}$ represents the same number of nodes in the $i$th community in $A$ and the $j$th community in $B$, $N$ represents the number of network nodes, and $L_{i.} (L_{.j})$ represents the $i$th row in $L$ (elements in column $j$).

## Synthetic and real network datasets

1) LFR network datasets

This article uses the LFR network model proposed by *Lancichinetti, Fortunato & Radicchi (2008)* to evaluate the performance of the proposed algorithm. The LFR model controls the generation of different network structures by adjusting different parameters. The parameter settings used in this article are shown in Table 2. Among them, the number of nodes is set to 100, 500, and 1,000 and represents three types of networks: LFR1, LFR2 and LFR3 respectively; $\mu$ controls the connectivity between communities and is set to 0.1 to 0.5; $O_n$ and $O_m$ represent the number of network overlapping nodes and the number of communities to which each overlapping node belongs; and represent the minimum number of nodes and the maximum number of nodes in each community respectively; $k$ and $k_{max}$ represent the average node degree and maximum node degree of the network respectively; $\tau_1$ and $\tau_2$ are the power law distribution indices.

**Table 2 LFR synthetic network datasets.**

| Network | $n$ | $\mu$ | $O_n$ | $O_m$ | $C_{min}$ | $C_{max}$ | Others |
|---------|-----|-------|-------|-------|-----------|-----------|--------|
| LFR1 | 100 | $\{0.1, 0.2, 0.3, 0.4, 0.5\}$ | $0.1n$ | $\{2, 4, 6\}$ | 10 | 20 | $k = 10$ |
| LFR2 | 500 | $\{0.1, 0.2, 0.3, 0.4, 0.5\}$ | $0.1n$ | $\{2, 4, 6\}$ | 30 | 60 | $k_{max} = 40$ |
| LFR3 | 1,000 | $\{0.1, 0.2, 0.3, 0.4, 0.5\}$ | $0.1n$ | $\{2, 4, 6\}$ | 30 | 60 | $\tau_1 = 2, \tau_2 = 1$ |

**Table 3 Real network datasets.**

| Network | Nodes | Edge | Avg degree | Community |
|---------|-------|------|------------|-----------|
| Karate[1] | 34 | 78 | 4.69 | 2 |
| Dolphin[2] | 62 | 159 | 5.13 | 2 |
| Football[3] | 115 | 613 | 10.66 | 12 |
| Polbook[4] | 105 | 441 | 8.4 | 3 |
| Email[5] | 1,133 | 5,451 | 4.81 | Unknown |
| Jazz[6] | 198 | 2,742 | 27.7 | Unknown |
| SFI[7] | 118 | 200 | 1.69 | Unknown |
| Y2H[8] | 1,966 | 2,705 | 2.75 | 203 |
| Yeast-D2[8] | 1,443 | 6,993 | 9.69 | 162 |

**Note:**
[1] http://konect.cc/networks/ucidata-zachary/
[2] http://konect.cc/networks/dolphins/
[3] http://konect.cc/networks/dimacs10-football/
[4] http://konect.cc/networks/dimacs10-polbooks/
[5] http://deim.urv.cat/alexandre.arenas/data/welcome.htm
[6] http://konect.cc/networks/arenas-jazz/
[7] https://doi.org/10.1073/pnas.122653799
[8] http://faculty.uaeu.ac.ae/nzaki/ProRank.htm.

2) Real network datasets

This article uses nine known real-world networks to evaluate algorithm performance. Six of these networks have real community structures. The specific information of the real network is shown in Table 3.

### Evaluation of non-overlapping community discovery experiments

As shown in Table 4, the modularity Q values obtained by the first-stage non-overlapping community discovery algorithm proposed in this article and the other three proposed non-overlapping community algorithms in seven real-world network datasets are given. It can be seen that the maximum modularity value of the non-overlapping communities in the first stage of the algorithm proposed in this article is the highest than that of other algorithms, and the average modularity value of only one data is lower than DECS. It shows that the first-stage method proposed in this article has better accuracy in non-overlapping community division. However, the single-objective optimization method proposed by CNLLP based on core nodes and layer-by-layer label propagation only considers the value of modularity and ignores the relationship with the real community, thus only obtaining a poor Q value. It also shows that simultaneous optimization of multiple objectives can yield better results than optimization of a single objective.

 

**Table 4 Statistical results of Q obtained through four methods on seven real-world networks.** The best results are shown in bold.

| Network | Indicators | DECS | EMOFM-DK | CNLLP | Proposed |
|---|---|---|---|---|---|
| Karate | Q_max | **0.4198** | **0.4198** | 0.3718 | **0.4198** |
| | Q_avg | **0.4198** | 0.4138 | 0.3718 | **0.4198** |
| | Std | 0.0000 | 0.0061 | 0.0000 | 0.0000 |
| Dolphin | Q_max | **0.5246** | 0.5151 | 0.4588 | **0.5246** |
| | Q_avg | **0.5245** | 0.5127 | 0.4588 | 0.5240 |
| | Std | 0.0001 | 0.0012 | 0.0000 | 0.0003 |
| Football | Q_max | **0.6044** | 0.6031 | 0.5214 | **0.6044** |
| | Q_avg | 0.6032 | 0.5923 | 0.5214 | **0.6038** |
| | Std | 0.0006 | 0.0054 | 0.0072 | 0.0003 |
| Polbook | Q_max | 0.5189 | 0.5201 | 0.4574 | **0.5213** |
| | Q_avg | 0.5144 | 0.5092 | 0.4574 | **0.5206** |
| | Std | 0.0023 | 0.0055 | 0.0000 | 0.0004 |
| Email | Q_max | 0.5001 | 0.4790 | 0.4938 | **0.5054** |
| | Q_avg | 0.4995 | 0.4615 | 0.4938 | **0.5030** |
| | Std | 0.0003 | 0.0087 | 0.0000 | 0.0012 |
| Jazz | Q_max | 0.4428 | 0.4233 | 0.4103 | **0.4436** |
| | Q_avg | 0.4342 | 0.40245 | 0.4103 | **0.4436** |
| | Std | 0.0043 | 0.0104 | 0.0000 | 0.0000 |
| SFI | Q_max | 0.742 | 0.7426 | 0.7098 | **0.7469** |
| | Q_avg | 0.7318 | 0.7326 | 0.7098 | **0.7468** |
| | Std | 0.0051 | 0.0051 | 0.0000 | 0.0000 |

When the LFR network is generated, two files will be obtained, including network data and the real community division structure of the network. Therefore, this article uses gNMI and EQ to evaluate the experimental results of the LFR network.

As shown in Figs. 4 to 6, the gNMI values obtained by 45 LFR networks on the four algorithms are shown. Taking LFR1 as an example, it shows the gNMI value detected when $\mu$ changes from 0.1 to 0.5 when node $N$ is 100 and $O_m$ is set to 2, 4, and 6. It can be seen from the LFR1, LFR2, and LFR3 network diagrams that the gNMI values of all algorithms almost show a decreasing trend as the $\mu$ value increases. For the gNMI value corresponding to each $\mu$, most of the algorithms proposed in this article are higher than The performance of the remaining algorithms is particularly obvious on the LFR3 network. As the number of nodes increases, the community results obtained by the proposed algorithm have greater advantages than other algorithms. Therefore, the algorithm proposed in this article has better performance and can search for a network structure closer to the real one.

As shown in Figs. 7 to 9, the EQ values obtained by the LFR1, LFR2, and LFR3 networks on the four algorithms are shown. In the LFR1 network result EQ diagram, when $O_m$ is 2 and 4 and $\mu$ is between 0.1 and 0.3, the EQ value obtained by the algorithm proposed in this article is almost equal to that of EMOFM-DK and is higher than the other two algorithms. At the same time, the EQ value of this algorithm when $O_m = 6$ is better than the

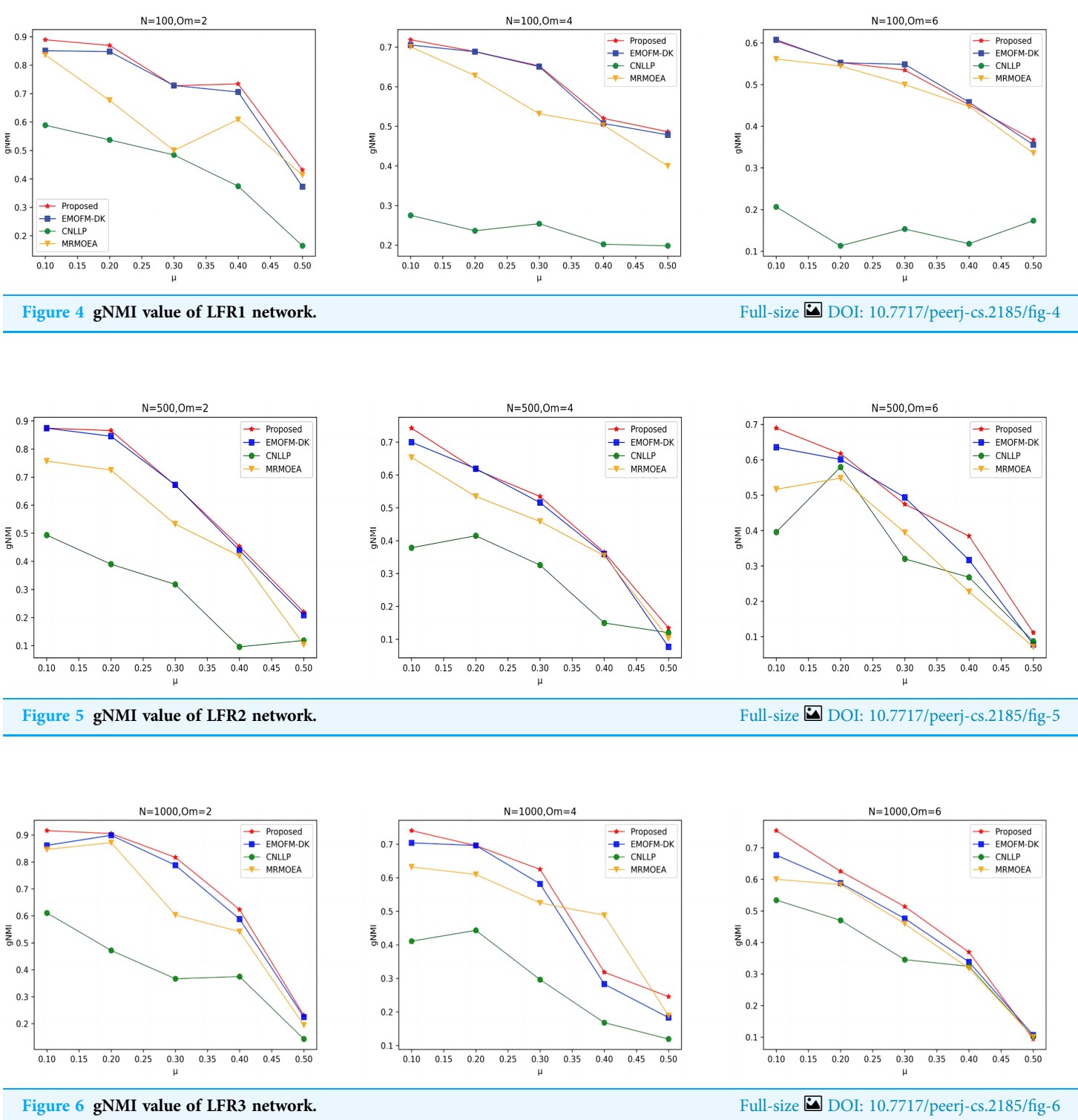

**Figure 4** gNMI value of LFR1 network.

**Figure 5** gNMI value of LFR2 network.

**Figure 6** gNMI value of LFR3 network.

other three algorithms. In addition, in the EQ diagram of the LFR2 and LFR3 network results, although the EQ value obtained by the algorithm proposed in this article is higher than the MRMOEA algorithm and the CNLLP algorithm, compared with the EMOFM-

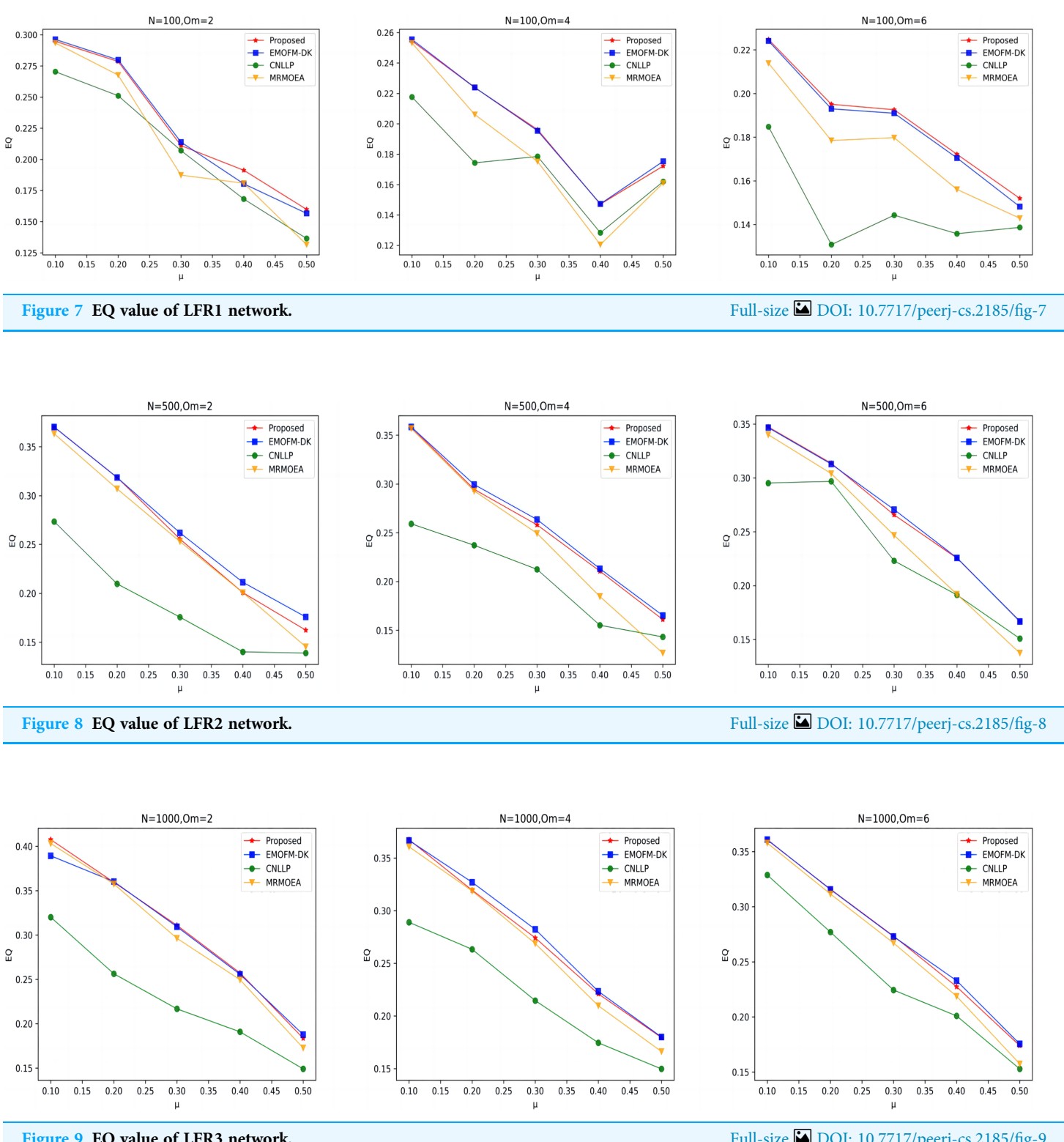

**Figure 7 EQ value of LFR1 network.**

**Figure 8 EQ value of LFR2 network.**

**Figure 9 EQ value of LFR3 network.**

**Table 5 Statistical results of EQ obtained through four methods on nine real-world networks.** The best results are shown in bold.

| Network | Indicators | MR-MOEA | EMOFM-DK | CNLLP | Proposed |
|---|---|---|---|---|---|
| Karate | EQ_max | 0.2285 | **0.2348** | 0.2108 | **0.2348** |
| | EQ_avg | 0.2285 | **0.2348** | 0.2108 | **0.2348** |
| | Std | 0.0000 | 0.0000 | 0.0000 | 0.0000 |
| Dolphin | EQ_max | 0.2646 | **0.2750** | 0.2401 | **0.2750** |
| | EQ_avg | 0.2623 | 0.2739 | 0.2401 | **0.2744** |
| | Std | 0.0011 | 0.0005 | 0.0000 | 0.0003 |
| Football | EQ_max | 0.3044 | 0.3066 | 0.2651 | **0.3067** |
| | EQ_avg | 0.3023 | 0.3066 | 0.2651 | **0.3067** |
| | Std | 0.0011 | 0.0000 | 0.0000 | 0.0000 |
| Polbook | EQ_max | 0.2641 | **0.2704** | 0.2354 | **0.2704** |
| | EQ_avg | 0.2628 | 0.2703 | 0.2354 | **0.2704** |
| | Std | 0.0007 | 0.0001 | 0.0000 | 0.0000 |
| Email | EQ_max | 0.2405 | 0.2737 | 0.1881 | **0.2830** |
| | EQ_avg | 0.2353 | 0.2725 | 0.1878 | **0.2801** |
| | Std | 0.0026 | 0.0006 | 0.0002 | 0.0014 |
| Jazz | EQ_max | 0.2190 | 0.2259 | 0.2087 | **0.2261** |
| | EQ_avg | 0.2164 | 0.2259 | 0.2087 | **0.2261** |
| | Std | 0.0013 | 0.0000 | 0.0000 | 0.0000 |
| SFI | EQ_max | 0.3705 | 0.3790 | 0.3646 | **0.3851** |
| | EQ_avg | 0.3688 | 0.3781 | 0.3646 | **0.3848** |
| | Std | 0.0009 | 0.0004 | 0.000 | 0.0001 |
| Y2H | EQ_max | 0.3213 | **0.3653** | 0.3150 | 0.3411 |
| | EQ_avg | 0.3206 | **0.3637** | 0.3149 | 0.3410 |
| | Std | 0.0003 | 0.0008 | 0.0001 | 0.0001 |
| Yeast-D2 | EQ_max | 0.4141 | **0.4168** | 0.3481 | 0.4144 |
| | EQ_avg | 0.4110 | **0.4162** | 0.3451 | 0.4140 |
| | Std | 0.0015 | 0.0003 | 0.0015 | 0.0002 |

DK algorithm, it is either equal to or lower than this algorithm. This is because in the multi-objective evolutionary algorithm, we use modularity EQ and gNMI as optimization goals, and the optimal Pareto values obtained conflict with each other, so a better gNMI value is obtained in the algorithm proposed in this article Afterwards, the resulting EQ value may drop.

## Real network experiment results

Table 5 and Fig. 10 shows the EQ values obtained by the algorithm proposed in this article and the other three algorithms in nine real-world network datasets. It can be seen that EMOFM-DK based on fuzzy clustering and the algorithm proposed in this article have obtained better modularity values on nine datasets, reflecting the better performance of the fuzzy clustering method in discovering overlapping nodes. CNLLP, which is based on core nodes and layer-by-layer label propagation, obtains poor results in dividing overlapping
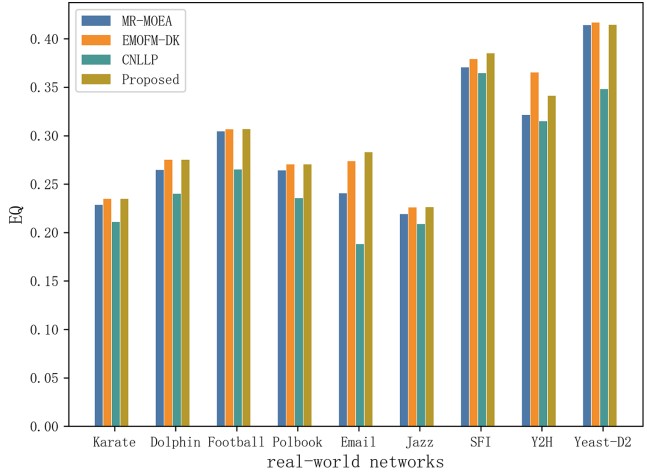

**Figure 10 Compare the EQ values of nine real network data on four algorithms.**

**Table 6 Statistical results of gNMI obtained through four methods on six real-world networks.** The best results are shown in bold.

| Network | Indicators | MR-MOEA | EMOFM-DK | CNLLP | Proposed |
|---|---|---|---|---|---|
| Karate | gNMI_max | **1.0000** | **1.0000** | 0.8372 | **1.0000** |
| | gNMI_avg | **1.0000** | **1.0000** | 0.8372 | **1.0000** |
| | Std | 0.0000 | 0.0000 | 0.0000 | 0.0000 |
| Dolphin | gNMI_max | **1.0000** | **1.0000** | 0.6011 | **1.0000** |
| | gNMI_avg | **1.0000** | **1.0000** | 0.6011 | **1.0000** |
| | Std | 0.0000 | 0.0000 | 0.0000 | 0.0000 |
| Football | gNMI_max | 0.8415 | 0.8370 | 0.6752 | **0.8545** |
| | gNMI_avg | 0.8067 | 0.8370 | 0.6752 | **0.8489** |
| | Std | 0.0174 | 0.0000 | 0.0000 | 0.0028 |
| Polbook | gNMI_max | 0.4561 | 0.3670 | 0.4829 | **0.5170** |
| | gNMI_avg | 0.4195 | 0.3553 | 0.4803 | **0.5118** |
| | Std | 0.0183 | 0.0058 | 0.0013 | 0.0026 |
| Y2H | gNMI_max | **0.1332** | 0.0930 | 0.0103 | 0.0925 |
| | gNMI_avg | **0.1316** | 0.0880 | 0.0089 | 0.0918 |
| | Std | 0.0008 | 0.0025 | 0.0007 | 0.0004 |
| Yeast-D2 | gNMI_max | 0.2170 | 0.2281 | 0.1796 | **0.2589** |
| | gNMI_avg | 0.2169 | 0.2280 | 0.1794 | **0.2567** |
| | Std | 0.0000 | 0.0001 | 0.0001 | 0.0011 |

communities. The reason may be that it only uses the degree of belonging of the node to determine whether the node is overlapping. In addition, on the nine datasets, the number of maximum and average values of EQ obtained by the algorithm proposed in this article is seven, while the maximum and average number of EQ obtained by EMOFM-DK are five and three respectively, indicating that the two-stage overlapping community discovery method proposed in this article can show better performance on the test data set.

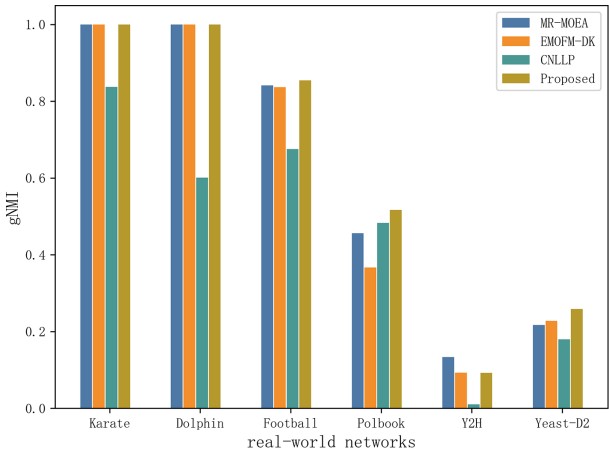

**Figure 11 Compare gNMI values of six real network data on four algorithms.**

Table 6 and Fig. 11 shows the gNMI values obtained by the algorithm proposed in this article and the other three algorithms in six real-world network datasets. It can be seen that on the two datasets of Karate and Dolphin, the algorithm proposed in this article and the two algorithms MR-MOEA and EMOFM-DK can find the real network partition structure. On the Y2H dataset, MR-MOEA achieved the best results. On the remaining three datasets, the optimal results obtained by the algorithm proposed in this article are higher than the other three algorithms. Therefore, the information feedback model proposed in this article divides overlapping community structures and uses values from historical evolutionary periods to optimize the current moment and obtain results closer to real communities.

## CONCLUSION AND FUTURE WORK

This article proposes a two-stage multi-objective evolutionary algorithm for overlapping community discovery problem. In the first stage, to obtain better non-overlapping communities, the algorithm first adopts a population initialization strategy based on central nodes, combining the maximum node degree and randomly generating central nodes to increase population diversity. Then, non-overlapping communities are optimized through a multi-objective evolution method to comprehensively consider multiple optimization objectives and improve community quality.

In the second stage, to discover better overlapping communities, the algorithm selects nodes from the non-overlapping communities obtained in the first stage as central nodes and uses the information feedback model to optimize the fuzzy threshold to more accurately identify overlapping nodes, thereby forming a more Suitable overlapping community.

Finally, comparative experiments are conducted on synthetic network datasets and real network datasets. Through experimental comparisons, the algorithm proposed in this article is evaluated in the first stage of non-overlapping community discovery and the second stage of overlapping community discovery algorithm compared with classic and

recent popular algorithms. It is proved that the algorithm proposed in this article has better performance on non-overlapping and overlapping community discovery algorithms.

In the process of network community segmentation, the growth of network data dynamically changes over time, and thus the structure of the community changes. Therefore, this article only applies to overlapping community discovery for static networks, where higher complexity may exist on large-scale networks. In the future, we will focus on exploring overlapping community discovery methods for dynamic and large-scale networks and developing more efficient search algorithms with lower time complexity.

### Funding
This work was supported by the National Natural Science Foundation of China (Grant No. 61862051), the Science and Technology Foundation of Guizhou Province (No. ZK[2022] 549), and the foundation of Qiannan Normal University for Nationalities (No. 2024ZDZK03). The funders had no role in study design, data collection and analysis, decision to publish, or preparation of the manuscript. The funders had no role in study design, data collection and analysis, decision to publish, or preparation of the manuscript.

### Grant Disclosures
The following grant information was disclosed by the authors:
National Natural Science Foundation of China: 61862051.
Science and Technology Foundation of Guizhou Province: ZK[2022]549.
Qiannan Normal University for Nationalities: 2024ZDZK03.

### Competing Interests
The authors declare that they have no competing interests.

### Author Contributions
- Lei Cai conceived and designed the experiments, performed the experiments, analyzed the data, performed the computation work, prepared figures and/or tables, authored or reviewed drafts of the article, and approved the final draft.
- Jincheng Zhou conceived and designed the experiments, analyzed the data, prepared figures and/or tables, authored or reviewed drafts of the article, and approved the final draft.
- Dan Wang conceived and designed the experiments, prepared figures and/or tables, authored or reviewed drafts of the article, and approved the final draft.

### Data Availability
The code and the datasets are available in the Supplemental Files.

## Supplemental Information

Supplemental information for this article can be found online at http://dx.doi.org/10.7717/peerj-cs.2185#supplemental-information.

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
