# Peer review of "Two-stage multi-objective evolutionary algorithm for overlapping community discovery"

_PeerJ Computer Science, doi:10.7717/peerj-cs.2185_

## Round 0.1 · original submission · Major Revisions

Dear authors,

Thank you for submitting your article. Feedback from the reviewers is now available. It is not recommended that your article be published in its current format. However, we strongly recommend that you address the issues raised by the reviewers, especially those related to readability, experimental design and validity, and resubmit your paper after making the necessary changes. When submitting the revised version of your article, it will be better to address the following:

1. The research gaps and contributions should be clearly summarized in the introduction section. Please evaluate how your study is different from others in the related work section.
2. Explanation of the equations should be checked. All variables should be written in italic as in the equations. Definitions of variables and the boundaries should be written. Some equations need appropriate referecences. They seem they are firstly proposed in this paper.
3. "Chapter" should be corrected as "Section".
4. The values for the parameters of the algorithms selected for comparison should be given.
5. The paper lacks the running environment, including software and hardware. The analysis and configurations of experiments should be presented in detail for reproducibility. It is convenient for other researchers to redo your experiments and this makes your work easy acceptance. A table with parameter settings for experimental results and analysis should be included in order to clearly describe them.
6. The authors should clarify the pros and cons of the methods. What are the limitation(s) methodology(ies) adopted in this work? Please indicate practical advantages, and discuss research limitations.
7. Please expand the future research directions.

Best wishes,

Reviewer 1 ·

Basic reporting

-There are minor errors in the writing language of the study, it should be reviewed.
- Studies on community discovery have been examined under different headings in the related studies section. If what is meant by "Tag-based propagation method" is "Label-based propagation method", this should be corrected.
- Abbreviations should be explained where they are first used (for example LPA)
- In the expression "between NCNi and NCNi" on page 10, line 302 of the article, is one of the NCNi's CNi?
-It should be noted that symbols and mathematical expressions are written in italics within the text.
- In the LFR network dataset section, spaces should be added between some expressions in lines 390, 391, 394, and 395, and spelling errors should be corrected to make the paper more readable.
- The statement on page 11, line 360 should be corrected.
- In this study, which proposes a two-stage multi-objective optimization method for overlapping community discovery, it would be better to add a flow diagram explaining the adaptation of the method to the problem.

Experimental design

- How was the value of 0.3 determined in Algorithm 1?
- It is stated that the fuzzy control parameter is generally set to 2. How does setting this value smaller or larger affect the performance of the algorithm?

Validity of the findings

- The motivation for the study should be given in the introduction section. It should be stated why the Evolutionary Algorithm is preferred.
- What is the 45 LFR network mentioned in the experimental results section? Information about 3 different types of LFR networks is given.
-How were optimal Pareto values used for the design of multi-objective optimization? It should be explained.

Reviewer 2 ·

Basic reporting

- This paper proposes a two-stage multi-objective evolutionary algorithm to solve the problem of overlapping community discovery.
- The proposed algorithm was evaluated and compared with other algorithms.
- The paper is well structured and organized.
- It is recommended to add additional recent references in the related work section with a focus on advantages and limitations of the discussed methods.

Experimental design

- The experiments were designed and conducted to evaluate the effectiveness of the proposed algorithm and compare it with other algorithms.
- Add a reference for the Generalized normalized mutual information (gNMI) metric that was used for evaluation.
- It will be helpful to provide more details about the real network datasets that were used in the experiments.

Validity of the findings

- A synthetic and real network datasets were used in these experiments.
- The proposed algorithm was evaluated in the first and the second stages.
- The proposed algorithm shows a better performance compared to other algorithms.
- It will be helpful to add additional graphs to represent the experimental results in tables 5 and 6.

Additional comments

- Equation 4 has three equations and they should be numbered separately (and then update the next numbering of the next equations)
- It will be helpful to add additional future directions for research in this area.

---

## Round 0.2 · accepted · Accept

Dear authors,

Thank you for the revision and for clearly addressing all the reviewers' comments. I confirm that the paper is improved. Your paper is now acceptable for publication in light of this revision.

Best wishes,

Reviewer 1 ·

Basic reporting

No comment

Experimental design

No comment

Validity of the findings

No comment

Additional comments

The authors made the relevant corrections, taking into account the evaluations.

Reviewer 2 ·

Basic reporting

The authors addressed the comments form the first review and made the required revisions.

Experimental design

The authors addressed the comments form the first review and made the required revisions.

Validity of the findings

The authors addressed the comments form the first review and made the required revisions.